# Gamification for the Improvement of Diet, Nutritional Habits, and Body Composition in Children and Adolescents: A Systematic Review and Meta-Analysis

**DOI:** 10.3390/nu13072478

**Published:** 2021-07-20

**Authors:** Nora Suleiman-Martos, Rubén A. García-Lara, María Begoña Martos-Cabrera, Luis Albendín-García, José Luis Romero-Béjar, Guillermo A. Cañadas-De la Fuente, José L. Gómez-Urquiza

**Affiliations:** 1Faculty of Health Sciences, University of Granada, Cortadura del Valle SN, 51001 Ceuta, Spain; norasm@ugr.es; 2Granada-Northeast Health Management Area, Andalusian Health Service, Ctra. de Murcia SN, 18800 Granada, Spain; ruben.garcia.lara.sspa@juntadeandalucia.es; 3Andalusian Health Service, San Cecilio Clinical University Hospital, Avenida del Conocimiento SN, 18016 Granada, Spain; mbmartos@ujaen.es; 4Andalusian Health Service, Granada-Metropolitan Health District, C/Joaquina Eguaras, 2, 18013 Granada, Spain; lualbgar1979@ugr.es; 5Department of Statistics and Operational Research, University of Granada, Av. Fuentenueva, 18071 Granada, Spain; 6Faculty of Health Sciences, University of Granada, Avenida de la Ilustración, 60, 18016 Granada, Spain; gacf@ugr.es (G.A.C.-D.l.F.); jlgurquiza@ugr.es (J.L.G.-U.)

**Keywords:** adolescents, children, dietary behaviour, game, gamification, healthy eating, nutrition

## Abstract

Currently, one of the main public health problems among children and adolescents is poor adherence to healthy habits, leading to increasingly high rates of obesity and the comorbidities that accompany obesity. Early interventions are necessary, and among them, the use of gamification can be an effective method. The objective was to analyse the effect of game-based interventions (gamification) for improving nutritional habits, knowledge, and changes in body composition. A systematic review and meta-analysis were performed in CINAHL, EMBASE, LILACS, MEDLINE, SciELO, and Scopus databases, following the PRISMA recommendations. There was no restriction by year of publication or language. Only randomized controlled trials were included. Twenty-three articles were found. After the intervention, the consumption of fruit and vegetables increased, as well as the knowledge on healthy food groups. The means difference showed a higher nutritional knowledge score in the intervention group 95% CI 0.88 (0.05–1.75). No significant effect of gamification was found for body mass index z-score. Gamification could be an effective method to improve nutritional knowledge about healthier nutritional habits. Promoting the development of effective educational tools to support learning related to nutrition is necessary in order to avoid and prevent chronic diseases.

## 1. Introduction

Nowadays, the absence of a physical exercise routine and adherence to a balanced diet are two major public health problems. In 2019, 38 million children under 5 years old were overweight or obese [1]. Furthermore, more than 80% of the adolescent population in the world does not do enough physical activity [2].

The WHO emphasizes that unhealthy diets and physical inactivity are two key risk factors to develop non-communicable diseases such as, cardiovascular diseases, cancer, and diabetes [3]. In addition, the intake of fruits and vegetables in the child population is under the recommended levels and that of sugar is well above the established limits, which increases the risk of developing these diseases [4].

Due to the lack of adherence to healthy habits in children and adolescents, a change in strategies focused on health promotion is required [5]. In this sense, gamification is a new, educative way that can be very useful to promote adherence to healthy habits [6].

Gamification is based on the application of game mechanics during the teaching–learning process [7]. In addition to using the intrinsic characteristics of a game, it also uses new technologies, the internet, and applications for mobile phones [8]. The characteristics of gaming are used to get achievements, prizes, or rewards [9]. This game dynamic is related to benefits at the learning level and increases the interaction between participants, offering freedom and increased motivation [10].

Gamification has been implemented in different areas of health and education. Benefits have been obtained in the improvement of healthy habits at the level of physical activity and nutrition in adults [7,11,12]. Gamification strategies have also been used in adolescents to improve sexual education [13,14] or to improve healthy habits, such as physical exercise and nutrition [15,16]. Many studies even focus on these interventions from an early age, for example, through interactive video games that improved physical activity [17,18,19], increased the number of daily steps [20,21], or improved cardiovascular parameters in the school population [22].

Due to the difficulty of inducing lifestyle changes in the young population, gamification can be an effective method to create change and improve adherence to healthy practices [6]. Some studies find that childhood is the ideal age to promote healthy habits, and resources such as social networks, mobile devices, or games can be very useful to promote knowledge and improve adherence [4].

Traditional interventions aimed at influencing fruit and vegetable intake among young people do not show extraordinary results [23]. There are few health applications that use gamification as an educational resource [24]. Given the need to carry out more innovative interventions focused on improving prevention strategies and policies, we performed this systematic review and meta-analysis in order to analyse behavioural changes in nutritional habits, knowledge, and body composition when using gamification as an educational resource.

The objective of this systematic review and meta-analysis was to analyse the effect of gamification for improving diet, nutritional habits, knowledge, and body composition in children and adolescents.

## 2. Methods

### 2.1. Design and Search Methods

A systematic review and meta-analysis was carried out following PRISMA (Preferred Reporting Items for Systematic Reviews and Meta-analyses) guidelines [25].

The following databases were consulted: CINAHL, EMBASE, LILACS, MEDLINE, SciELO, and Scopus. The search was carried out in April 2021. There was no restriction by year of publication. Using the Mesh terms, the search equation was “(game OR gamification) AND (child OR adolescent) AND (nutrition OR feeding behaviour OR food OR diet OR body composition OR body mass index OR health) AND (RCT OR randomized controlled trial)”.

For the selection of the study sample, the PICO (population, intervention, comparison, and outcome) strategy was used. The population were children and adolescents; the intervention was through different gamification programs (defined as organized games with a set of rules for playing and achieving goals or objectives by providing feedback and interaction); the comparison was made between the pre- and post-intervention groups, or the intervention and control groups; and the outcomes were diet and body composition improvement.

### 2.2. Inclusion and Exclusion Criteria

The studies with the following characteristics were included: (1) randomized controlled trials, (2) study sample comprising children and adolescents, (3) intervention as a playful game component, (4) gathering data on the effect of the intervention on eating habits, knowledge, and body composition. There was no restriction by language or by year of publication.

The exclusion criteria were (1) studies related to improvement in other health habits, (2) interventions applied exclusively to certain diseases/pathologies, (3) combination with other types of interventions such as physical activity, (4) studies without a control group.

### 2.3. Study Selection, Quality Appraisal, and Risk of Bias

The selection was carried out by two authors independently. First, the title and abstract were read. Then the full text was read. Finally, a critical reading of the selected studies was carried out to assess the risk of bias. A third author was consulted in case of disagreement.

The quality of the studies was assessed following the levels of evidence and grades of recommendation of the Oxford Centre for Evidence-Based Medicine (OCEBM) (Howick et al., 2011) (Table 1). Risk of bias was analysed by pairs of independent reviewers using the Cochrane Collaboration risk of bias tool [26]. All the articles reached a quality level according to the quality assessment tools; therefore, no study was excluded.

### 2.4. Data Abstraction

A descriptive analysis was performed to extract the data from each included study, consulting with a third author in case of disagreement. The variables obtained from the selected articles were (1) author, year of publication, country, (2) design, (3) sample size, (4) aim, (5) type of intervention and duration, (6) measuring instruments, and (7) main results.

The coding reliability was calculated according to the intraclass correlation coefficient, which gave a mean value of 0.96 (minimum = 0.92, maximum = 1), and Cohen’s kappa coefficient, mean value 0.93 (minimum = 0.92, maximum = 1).

### 2.5. Data Analysis

The meta-analysis compared the standardized means difference between the gamification group and the control group. The heterogeneity of the sample was assessed with the I^2^ index, if this was greater than 50% a random-effects analysis was selected [26]. Publication bias was assessed using the funnel plot, and a sensitivity analysis was also performed. RevMan Web software was used.

## 3. Results

### 3.1. Study Characteristics

The database search comprised a total of 1433 articles. The final sample was *n* = 23 articles. The selection process is shown in Figure 1.

All included studies were clinical trials [27,28,29,30,31,32,33,34,35,36,37,38,39,40,41,42,43,44,45,46,47,48,49]. The total sample was 11,280 children and adolescents. The oldest article dates from 2005, although most studies were published from 2010 (*n* = 20). Most of the studies were conducted in the USA (*n* = 8) and Italy (*n* = 4). The adherence rate to the intervention programs was high, from 96.4% [40] to 91% [42,46]. The main characteristics of all the included studies are listed in Table 1.

### 3.2. Effects of Gamification on Food Groups and Food Habits

Providing nutritional information through gamification interventions had a positive influence on food selection in children and adolescents. After the intervention, the consumption of certain food groups increased. Weekly intake of fruit [28,35,39,49] and vegetables [27,33,35,39,49] was augmented with an increase of about 0.67 servings per day up to 1 serving [30,32]. Intake of nutrients related to fruits and vegetables such as vitamin C, beta-carotene, potassium, and dietary fibre improved [31,46]. The consumption of whole and protein-rich foods was also increased [35], and the sugar intake decreased significantly [42]. Some authors when analysing water consumption did not find significant differences [30] although they increased motivation [39].

Knowledge about information related to food groups increased from 4.8% [29] up to 34.10% [36]. In addition, knowledge about the five major macronutrients improved after intervention, although not at follow-up [37]. It also improved self-efficacy in the adoption of healthy eating practices such as the preparation of healthy meals [31].

Regarding eating habits, after the intervention, the frequency of eating while watching television or studying as well as eating in fast food restaurants was reduced [31].

### 3.3. Effects of Gamification in Body Composition

Regarding changes in body composition, some authors found significantly lower changes after intervention in waist circumference and BMI z-score after intervention and follow-up [47,48,49]. Although other authors did not find significant differences after intervention [27,30].

### 3.4. Meta-Analysis Results and Risk of Bias

There was no publication bias, and no studies were removed after sensitivity analysis. Regarding nutritional knowledge variable, six studies provided the necessary data, with a final sample of *n* = 2574 subjects in the intervention group and *n* = 2649 in the control group. The standardized means difference, with the 95% confidence interval, was 0.88 (0.05–1.75) and displayed a statistically significant higher knowledge score in the values of the experimental group (*p* < 0.05). These data mean that using gamification helps to improve nutritional knowledge about healthier nutritional habits. Although, in real clinical practice, that difference was not large enough to be statistically significant. Analysing BMI z-score, only two studies had the necessary information for the meta-analysis and showed no significant effect of gamification in this outcome. The forest plot and the risk of bias of each study are shown in Figure 2 and Figure 3.

## 4. Discussion

To our knowledge, this is the first systematic review and meta-analysis that analyses the benefits of interventions based on the improvement of nutritional habits, knowledge, and changes in body composition in children and adolescents.

Game-based interventions showed improvements in the selection of healthy food groups within this population. As indicated in other studies, the consumption of fruits and vegetables increased [50,51,52,53,54,55]. Although other studies also found positive effects in knowledge about food groups, these were not reflected in an increase in the intake of this food group [53,56].

Knowledge about the five macronutrient groups also increased, as noted by other authors, although without maintained effects [57]. Other studies after online game interventions in nutritional programs showed improvements in calcium and vitamin D intake [58]. In addition, the frequency and quantity of sugar-sweetened beverage intake were also reduced [59]; while water intake, as corroborated by other studies, did not improve [59].

The gamification of nutrition can lead to improvements in dietary behaviour among adolescents in the short term [60]. Some studies that used card games found a 23.1% reduction in the number of students who did not eat breakfast, maintaining this habit up to 3 months later [61]. Others even found an improvement in adherence to the Mediterranean diet [54].

Several studies showed that gamification was effective in teaching nutrition and weight management knowledge, as well as in the intention to follow a healthy diet [62,63]. However, in this study, there were few articles that analysed changes in body composition, although the results found showed benefits in waist circumference and BMI. Similar studies conducted in children with obesity and pre-type II diabetes after mobile application interventions found improvements in BMI, waist circumference, and percentage of body fat maintained over time [64], and even in adults they found improved weight, BMI, fat mass, waist circumference, and cholesterol [63].

The meta-analysis showed a higher nutritional knowledge score after the intervention, as pointed out by other authors [65,66,67] and that the psychoeducational multimedia games had the potential to substantially change dietary behaviour [68].

Participants preferred to select healthier foods after playing [66]. At these ages, game-based interventions could exert a very positive influence in improving health. Through gamification, improvements have been made in sexual education [13] and smoking habits [69]. In addition, these strategies have also shown benefits in teaching processes in students of different educational levels [10,70].

Gamification was a useful method for improving health habits. The gamification process gives participants the possibility to learn and face the challenge through a different and exciting process that allows them to increase the degree of commitment [8]. In addition, motivation increases thanks to obtaining prizes and rewards [71,72]. All of this improved self-efficacy to select healthy foods and adherence to healthy lifestyles [73].

According to current international standards, the child and adolescent population eat insufficient fruit and vegetables and a lot of processed foods with added sugars [74]. The healthy eating habits that are acquired in childhood are maintained in adulthood, so it is essential to promote an adequate intake. Given that more traditional health interventions have limited success, health policies could focus on the implementation of gamification projects in the school environment.

### Limitations

This study has several limitations. First, although all studies use gamification as an intervention, the great variability in the duration of the intervention and minutes of play may influence the heterogeneity of the results. On the other hand, the interventions were relatively short in time. Furthermore, the sample size of the clinical trials was small and the monitoring of the effect of the intervention over time was little studied. Finally, it was not possible to perform a meta-analysis about important outcomes such as fruit/vegetable intake, due to the heterogeneity in the units of measurement and presentation of the studies’ data.

Future research would be necessary to analyse the improvement in all food groups by age, as well as the changes in body parameters. Furthermore, more clinical trials would be necessary to determine the lasting effects of the enhancement over time.

## 5. Conclusions

Gamification was a positive influence on dietary behaviour and nutritional knowledge. The choice of food groups improved, highlighting an increase in the consumption of fruit and vegetables. Furthermore, the results of the meta-analysis showed an increase in the level of nutritional knowledge, but a significant effect of gamification was not found for body mass index z-score. Game-based interventions could be very helpful in promoting healthy habits. Promotion of the development of effective educational tools to support children in nutrition learning is necessary in order to avoid and prevent chronic diseases related to nutrition from childhood.

## Figures and Tables

**Figure 1 nutrients-13-02478-f001:**
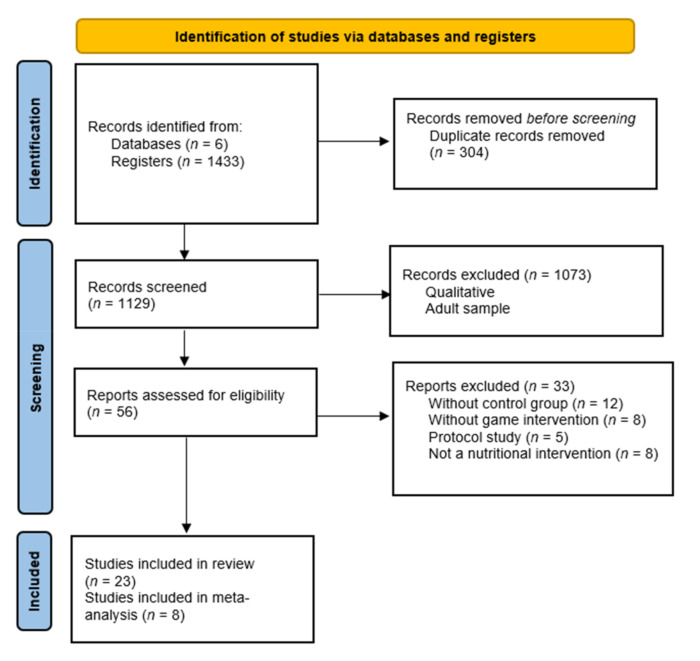
Flow diagram of the publication search process.

**Figure 2 nutrients-13-02478-f002:**
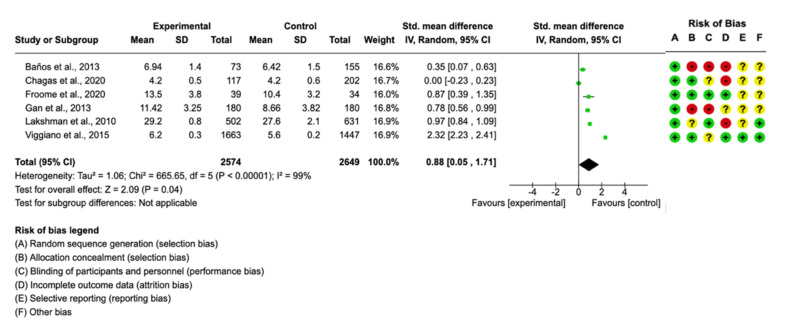
Forest plot for nutritional knowledge.

**Figure 3 nutrients-13-02478-f003:**
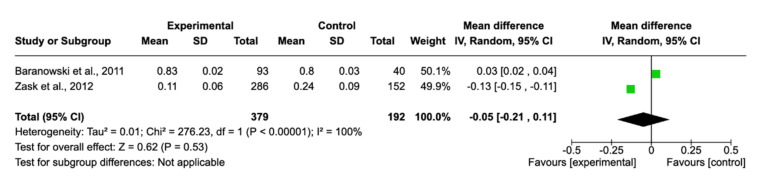
Forest plot for body mass index z-score.

**Table 1 nutrients-13-02478-t001:** Characteristics of the included studies *(n* = 23).

Authors, (Year), Country	Design	Sample	Aim	Intervention	Duration	Measurement	Main Outcomes M(SD)	EL/RG
Amaro et al. [27] (2006), Italy	RCT	*N* = 241 children Age 11–14 year *n* CG = 88 *n* IG = 153	To test the changes in nutrition knowledge and dietary behaviour	CG: no intervention IG: “Kalèdo” Nutrition board-game (play session 15–30 min)	24 weeks	Questionnaires on nutritional knowledge and food intake BMI	*Nutrition knowledge* Significant difference between IG and CG at post-assessment (*p* < 0.05). Adjusted means were 11.24 (95% CI 10.68–11.80) for the IG and 9.24 (95% CI 8.50–9.98) for CG *Dietary intake* Significant difference between IG and CG at post-assessment (*p* < 0.01) for the variable vegetable intake. Adjusted mean number of servings per week was 3.7 (95% CI 3.5–4.1) for IG and 2.8 (95% CI 2.4–3.3) for CG *BMI* No significant difference between IG and CG at post-assessment. Adjusted means were 0.345 (95% CI 0.29–0.39) for IG and 0.405 (95% CI 0.34–0.46) for CG	1b/A
Bannon et al. [28] (2006), USA	RCT	*N* = 50 children Mean age 5 years *n* CG = 18 *n* IG1 = 14 *n* IG2 = 18	To test the influence of nutrition message framing on snack choice among children	CG: control video IG1: gain-framed nutrition video message IG2: loss-framed nutrition video message	60 s video time	Food preference questionnaire Healthy food questionnaire	Gain- and loss-framed messages promoting healthy snacks have the potential to positively influence children’s behaviour In IG, 56% chose apples rather than animal crackers; in CG, only 33% chose apples	1b/A
Baños et al. [29] (2013), Spain	RCT	*N* = 228 children Age 10–13 years *n* CG = 155 *n* IG = 73	To study an online game to improve children’s nutritional knowledge	CG: paper–pencil intervention IG: “ETIOBE Mates” broader e-therapy platform educational website including serious games	2 weeks (unlimited sessions)	Nutritional knowledge questionnaire	**Baseline** *Nutritional knowledge* CG: 6.25 (1.3) IG: 6.46 (1.3) **2 weeks follow-up** *Nutritional knowledge* CG: 6.42 (1.5) IG: 6.94 (1.4)	1b/A
Baranowski et al. [30] (2011), USA	RCT	*N* = 133 children Age 10–12 years *n* CG = 40 *n* IG = 93	To promote behaviour change on children’s diet	CG: games on popular websites IG: *“*Escape from Diab” + “Nanoswarm: Invasion from Inner Space” Video-games (9 sessions of 40 min)	2 months	BMI Mean levels of FV, non-fat vegetables, total energy	**Baseline** **CG** *Body Composition* BMI %: 75.12 (1.04) BMI z-score: 0.78 (0.03) *Diet (Servings)* FV: 1.56 (0.18) Total Energy (kcal): 1657 (58) **IG** Body Composition BMI %: 77.41 (0.74) BMI z-score: 0.85 (0.02) *Diet (Servings)* FV: 1.88 (0.13) Total Energy (kcal): 1604 (41) **2 months follow-up** **CG** *Body Composition* BMI %: 75.98 (1.09) BMI z-score: 0.80 (0.03) *Diet (Servings)* FV: 1.48 (0.19) Total Energy (kcal): 1653 (63) **IG** *Body Composition* BMI %: 77.28 (0.75) BMI z-score: 0.83 (0.02) *Diet (Servings)* FV: 2.15 (0.13) Total Energy (kcal): 1632 (42)	1b/A
Chagas et al. [31] (2020), Brazil	RCT	*N* = 319 adolescents Mean age 15.8 years *n* CG = 202 *n* IG = 117	To assess the impact of a game-based nutritional intervention on food consumption, nutritional knowledge, and self-efficacy	CG: no intervention IG: “Rango Cards”, a digital game (unlimited session)	17 days	Nutritional knowledge questionnaire	**Baseline** *Nutritional knowledge* CG: 4.3 (0.5) IG: 4.2 (0.4) **Follow-up** *Nutritional knowledge* CG: 4.2 (0.6) IG: 4.2 (0.5)	1b/A
Cullen et al. [32] (2005), USA	RCT	*N* = 1489 children Age 8–12 years *n* CG = 740 *n* IG = 749	To assess changes in nutritional practices	CG: no intervention IG: “Squire’s Quest!” multimedia game (10 sessions of 25 min)	5 weeks	Servings of fruit, 100% juice, and vegetables consumed	After intervention, at snacks, the difference in means between IG and CG was significantly higher for fruit and 100% fruit juice, and for regular non-fried vegetables but not for other juice and vegetables	1b/A
Cullen et al. [33] (2016), USA	RCT	*N* = 387 children Age 9–11 years *n* CG = 97 *n* IG1 action plans = 98 *n* IG2 coping plans = 95 *n* IG3 action + coping plans = 97	To examine an online video-game to promote fruit-vegetable consumption changes	CG: no intervention IG: “Squire’s Quest II” online video-game. 10 sessions (25 min each) for 5 weeks	5 weeks 6 months follow up	Servings of fruit, 100% juice, and vegetables consumed	At 6 months, vegetable intake at dinner was significantly increased in action and coping groups. Overall, there were significant increases in fruit consumption at breakfast (*p* = 0.009), lunch (*p* = 0.014), and as a snack (*p* < 0.001)	1b/A
Folkvord et al. [34] (2013), Netherlands	RCT	*N* = 270 children Age 8–10 years *n* CG = 69 *n* IG1 = 69 *n* IG2 = 67 *n* IG3 = 65	To examine the effect of advergames that promote intake of energy-dense snacks or fruit on children	CG: no intervention IG1: advergame that promoted energy-dense snacks IG1: advergame that promoted fruit IG3: non-food products	-	Caloric intake	Children who played an advergame that promoted food (energy-dense snacks or fruit) ate significantly more than did the children who played an advergame that promoted non-food products (*p* < 0.01) and also ate. Sex (male) (*p* < 0.05), hunger (*p* < 0.01), and age (*p* < 0.05) were significantly related to energy-dense calorie intake	1b/A
Froome et al. [35] (2020), Canada	RCT	*N* = 73 children Age 8–10 years *n* CG = 34 *n* IG = 39	To determine improvement in children’s nutritional knowledge	CG: cooking game “My Salad Shop Bar” IG: game mobile application Foodbot Factory (learning module of drinks, whole-grain food, vegetables and fruits, animal protein, plant-based protein + voiceover) (10–15 min each day)	5 days	Nutrition knowledge	**Baseline** *Nutrition knowledge* CG: 10.2 (3.1) IG: 10.3 (2.9) **Day 5** *Nutrition knowledge* CG: 10.4 (3.2) IG: 13.5 (3.8)	1b/A
Gan et al. [36] (2019), Philippines	RCT	*N* = 360 children Age 7–10 years *n* CG = 180 *n* IG = 180	To increase nutrition knowledge	CG: no intervention IG: “Healthy Foodie” nutrition game (25 to 40 min to complete the game)	2 weeks	Nutrition knowledge questionnaire	**Baseline** *Food Group Knowledge score* CG: 9.55 (3.72) IG: 9.08 (3.48) *Food Frequency Knowledge score* CG: 9.67 (2.79) IG: 9.16 (2.55) **2 weeks follow-up** *Food Group Knowledge score* CG: 8.66 (3.82) IG: 11.42 (3.25) *Food Frequency Knowledge score* CG: 9.22 (2.75) IG: 10.55 (2.28)	1b/A
Hermans et al. [37] (2018), USA	RCT	*N* = 108 children Age 10–13 years *n* CG = 58 *n* IG = 50	To test the short-term effectiveness of a videogame designed to teach elementary school children about nutrition and healthy food choices	CG: web-based nutrition game “Super Shopper” (not designed to educate children in healthy food choices) IG: gameplay of An Alien Health Game “Feed the Alien!” (designed to educate children in healthy food choices and the main function of the five most important macronutrients). 1 h of gameplay (30 min session)	2 weeks	Nutritional knowledge Food intake	*Nutritional knowledge* IG had better knowledge at immediate post-test, (*p* < 0.001) but not at 2-week follow-up (*p* = 0.999) *Food intake* Participants in both conditions ate more energy-dense foods at 2-week follow-up than at immediate post-test (*p* < 0.001).	1b/A
Lakshman et al. [38] (2010), UK	RCT	*N* = 1133 children Age 9–11 years *n* CG = 631 *n* IG = 502	To increase nutrition knowledge	CG: traditional healthy eating curriculum IG: “Top Grub”: card nutrition game	9 weeks	Nutrition knowledge questionnaire Attitudes to healthy eating	**Baseline** *Nutrition knowledge Total score* CG: 27.3 (2) IG: 28.3 (1.1) *Balanced diet domain (max 0.15 points)* CG: 11.3 (0.9) IG: 11.6 (0.4) *Ability to identify healthier foods* CG: 11.6 (0.9) IG: 12.1 (0.6) **9 weeks** Nutrition knowledge *Total score* CG: 27.6 (2.1) IG: 29.2 (0.8) *Balanced diet domain (max 0.15 points)* CG: 11.5 (0.9) IG: 12.1 (0.5) *Ability to identify healthier foods* CG: 11.6 (1.0) IG: 12.1 (0.4)	
Lu et al. [39] (2012), USA	RCT	*N* = 153 children Age 10–12 years *n* CG = 50 *n* IG = 103	To analyse positive health outcomes	CG: no intervention IG: health videogame “Escape from Diab”. 9 sessions of 40 min	2 months	Fruit, vegetables, and water consumption	**Baseline vs. 2 months follow-up** (Only for IG) Fruit/Vegetables Preference 68.36 (13.53)/71.54 (15.49) Water Preference 2.64 (0.65)/2.59 (0.72) Intrinsic Motivation for Fruit 5.89 (1.94)/6.15 (2.18) Intrinsic Motivation for Vegetable 3.76 (1.82)/3.73 (1.94) Intrinsic Motivation for Water 5.19 (1.95)/5.51 (1.91) Fruit Self-Efficacy 9.49 (2.12)/10.39 (2.29) Vegetable Self-Efficacy 4.69 (2.24)/5.32 (2.22) Water Self-Efficacy 3.56 (1.39)/3.69 (1.54) Story immersion correlated positively (*p* < 0.03) with an increase in Fruit and Vegetable Preference (r = 0.27), Intrinsic Motivation for Water (r = 0.29), Vegetable Self-Efficacy (r = 0.24)	1b/A
Mack et al. [40] (2020), Germany	RCT	*N* = 82 children Age 9–12 year *n* CG = 40 *n* IG = 42	To evaluate the gain in knowledge about important lifestyle factors with the focus on nutrition	CG: brochure healthy lifestyle IG: nutrition games modules (2 sessions of 45 min)	2 weeks	Maintenance of knowledge questionnaire Food frequency questionnaire Healthy nutrition index	**Baseline** *Knowledge score* *% Food pyramid score* CG: 49 (14) IG: 50 (13) *% of dietary energy-density score* CG: 41 (19) IG: 51 (18) *Healthy nutrition index (reported by children)* CG: 8.9 (2.2) IG: 8.8 (2.1) **4 weeks follow-up** *Knowledge score**% Food pyramid* CG: 54 (12) IG: 77 (12) *% of dietary energy-density score* CG: 46 (22) IG: 64 (17) *Healthy nutrition index (reported by children)* CG: 9.3 (2.5) IG: 9.5 (2.2)	1b/A
Putnam et al. [41] (2018), USA	RCT	*N* = 132 children Age 4–5 years *n* CG = 44 *n* IG = 88	To encourage healthier snack selection and consumption	CG: game adventure app IG: game adventure app with “Dora the Explorer”	-	Snack choices	Children who were aware of Dora were 10.34 times more likely to select healthier snack items than those who were unaware of her (*p* = 0.008)	1b/A
Sharma [42] (2015), USA	RCT	*N* = 94 children Age 8–12 years *n* CG = 50 *n* IG = 44	To evaluate dietary behaviours	CG: usual programs IG: “Quest to Lava Mountain” adventure game (90 min play game)	6 weeks	Dietary Intake	**Baseline** *Dietary intake fruit (servings per 1000 kcal)* CG: 0.81 (0.67) IG: 0.84 (0.67) *Vegetables (servings per 1000 kcal)* CG: 0.51 (0.33) IG: 0.56 (0.42) *Dietary fibre (g/1000 kcal)* CG: 8.82 (2.46) IG: 8.29 (2.59) *Sugars (g/1000 kcal)* CG: 55.33 (16.94) IG: 55.35 (13.47) *Total fat (g/1000 kcal)* CG: 32.31 (6.01) IG: 32.84 (5.45) *Energy (kcal)* CG: 1632.51 (443.37) IG: 1415.49 (412.02) *Carbohydrates (g/1000 kcal)* CG: 51.83 (7.97) IG: 49.79 (6.98) *Protein (g/1000 kcal)* CG: 15.86 (3.71) IG: 17.37 (3.80) *Calcium (mg/1000 kcal)* CG: 520.92 (180.32) IG: 597.36 (186.07) **6 weeks follow-up** *Dietary intake fruit (servings per 1000 kcal)* CG: 0.79 (0.68) IG: 0.71(0.67) *Vegetables (servings per 1000 kcal)* CG: 0.45 (0.37) IG: 0.50 (0.44) *Dietary fibre (g/1000 kcal)* CG: 7.96 (2.82) IG: 8.35 (2.61) *Sugars (g/1000 kcal)* CG: 60.94 (15.97) IG: 50.45 (18.93) *Total fat (g/1000 kcal)* CG: 31.90 (6.83) IG: 34.78 (6.83) *Energy (kcal)* CG: 1331.46 (524.92) IG: 1304.11 (571.60) *Carbohydrates (g/1000 kcal)* CG: 52.40 (8.31) IG: 48.49 (9.03) *Protein (g/1000 kcal)* CG: 15.70 (4.21) IG: 16.72 (5.72) *Calcium (mg/1000 kcal)* CG: 561.83 (262.32) IG: 538.15 (168.55)	1b/A
Sharps et al. [43] (2016), UK	RCT	*N* = 143 children Age 6–11 years *n* CG = 46 *n* IG1 = 49 *n* IG2 = 48	To increase intake of fruit and vegetables through board games	CG: non-food-related game IG1: descriptive social norm-based message. Fruit and vegetables related game IG2: health message and image condition. Fruit and vegetables related game 7 min of playtime every day	-	Food intake	*Food intake* Significant main effect of condition on fruit and vegetable intake (*p* = 0.01). IG2 ate significantly more fruit and vegetables than children in CG (*p* = 0.009). There was no significant main effect of conditioning on high-calorie snack food intake (*p* = 0.99)	1b/A
Rosi et al. [44] (2016), Italy	RCT	*N* = 145 children Age 8–10 years *n* CG = 33 *n* IG1 = 58 *n* IG2 = 54	To improve nutritional education	CG: no intervention IG1: “Master of Taste” nutritional educator IG2: “Master of Taste” supported by a humanoid robot	1 year	Cultural–nutritional awareness factor (score of the nutritional knowledge level)	**Baseline** *Cultural–nutritional awareness factor* CG: 5.5 (1.5) IG1: 6.2 (1.7) IG2: 5.9 (1.3) **1 year follow-up** *Cultural–nutritional awareness factor* CG: 6.1 (1.4) IG1: 6.9 (1.1) IG2: 6.9 (1.1)	1b/A
Spook et al. [45] (2016), Netherlands	RCT	*N* = 231 adolescents Mean age 17.28 years *n* CG = 126 *n* IG = 105	To assess dietary intake	CG: no intervention IG: “Balance It”, interactive multimedia game (unlimited sessions)	4 weeks	Dietary intake (fruit and vegetable consumption, snack consumption, and soft drink consumption)	**Baseline** *Behavioural outcomes* *(mean portion/day)* *Fruit intake* CG: 0.80 (0.68) IG: 0.81 (0.68) *Vegetable intake* CG: 1.32 (0.38) IG: 1.26 (0.33) *Snack consumption* CG: 0.98 (0.51) IG: 0.91 (0.50) *Soft drink consumption* CG: 1.11 (0.59) IG: 1.07 (0.53) **4 weeks follow-up** *Behavioural outcomes* (mean portion/day) *Fruit intake* CG: 0.81 (0.62) IG: 1.05 (0.75) *Vegetable intake* CG: 1.28 (0.36) IG: 1.21 (0.41) *Snack consumption* CG: 0.90 (0.48) IG: 0.86 (0.51) *Soft drink consumption* CG: 1.07 (0.57) IG: 0.92 (0.57)	1b/A
Thompson et al. [46], (2016) USA	RCT	*N* = 387 children Aged 9–11 years *n* CG = 97 *n* IG1 action = 98 *n* IG2 coping = 95 *n* IG3 action + coping = 97	To evaluate the dietary intake of healthy children	CG: no intervention Intervention: serious game “Squire’s Quest! II” (10 sessions of 25 min) IG1 Action: set a goal and then created an action plan to meet the goal. IG2: Coping, a goal to eat more FV and then to create a coping plan IG3: Both IG1 + IG2	5 weeks	Fruit and vegetable intake	***Baseline*** *Energy (kcal)* CG: 1496 (34.71) IG1: 1477 (34.93) IG2: 1487 (35.04) IG3: 1476 (35.13) *Vitamin C (mg)* CG: 96.89 (7.97) IG1: 74.25 (8.01) IG2: 73.37 (8.03) IG3: 84.99 (8.04) *Sodium (mg)* CG: 2655 (45.33) IG1:2626 (45.58) IG2: 2646 (45.71) IG3: 2623 (45.81) *Potassium (mg)* CG: 1732 (38.1) IG1: 1668 (38.35) IG2: 1693 (38.48) IG3: 1823 (38.59) *Total dietary fibre (g)* CG: 11.02 (0.34) IG1: 11.33 (0.35) IG2: 11.16 (0.35) IG3: 11.4 (0.35) *Added sugars (g)* CG: 54.74 (2.25) IG1: 58.69 (2.26) IG2: 56.83 (2.26) IG3: 58.16 (2.27) **6 months follow-up** *Energy (kcal)* CG: 1523 (39.09) IG1: 1444 (38.94) IG2: 1510 (39.79) IG3: 1482 (39.31) *Vitamin C (mg)* CG: 92.66 (11.98) IG1: 87.22 (11.91) IG2: 96.91 (12.11) IG3: 104.47 (11.98) *Sodium (mg)* CG: 2740 (50.96) IG1: 2562 (50.70) IG2: 2667 (51.91) IG3: 2670 (51.14) *Potassium (mg)* CG: 1789 (45.66) IG1: 1905 (45.47) IG2: 1854 (46.42) IG3: 1858 (45.86) *Total dietary fibre (g)* CG: 11.5 (0.42) IG1: 12.75 (0.42) IG2: 12.14 (0.42) IG3: 12.15 (0.42) Added sugars (g) CG: 48.2 (2.25) IG1: 50.79 (2.24) IG2: 53.83 (2.29) IG3: 50.13 (2.26)	1b/A
Viggiano et al. [47] (2015), Italy	RCT	*N* = 3110 children Age 9–19 years *n* CG = 1447 *n* IG = 1663	To promote nutrition education and to improve dietary behaviour	CG: no intervention IG: “Kalèdo” nutrition board-game (15–30 min session)	20 weeks	BMI z-score Nutrition Knowledge	**Baseline** *Normal Weight* CG: 52.55% IG: 51.6% *Overweight* CG: 32.6% IG: 34.9% *Obesity* CG: 14.8% IG: 13.3% *BMI z-score* CG: 0.59 IG: 0.58 *Nutrition Knowledge* CG: 4.4 (4.2–4.5) IG: 4.2 (4.1–4.4) *Food Habits* CG: 27.3 IG: 27.2 **18 months follow-up** *Normal Weight* CG: 52.55% IG: 63.55% *Overweight* CG: 32% IG: 27.8% *Obesity* CG: 15.45% IG: 8.65% *BMI z-score* CG: 0.58 IG: 0.34 *Nutrition Knowledge* CG: 5.6 (5.4–5.7) IG: 6.2 (6.1–6.4) *Food Habits* CG: 28.6 IG: 29.3	1b/A
Viggiano et al. [48] (2018), Italy	RCT	*N* = 1007 children Age 7–11 years *n* CG: 356 *n* IG: 651	To improve knowledge in nutrition and to promote a healthy lifestyle	CG: no intervention IG: “Kalèdo” nutrition board-game. 20 sessions of 15–30 min	20 weeks	Food frequency consumption BMI z-score	**8 months follow-up** IG significantly increased the consumption of healthy food (*p* < 0.01) compared to CG **18 months follow-up** The increase in the consumption of healthy foods in GI was maintained over time (*p* < 0.01). Significantly higher consumption of healthy food in girls (mean 9.41; CI 95% 7.61–11.22) compared to boys (mean 7.11; CI 95% 5.46–8.76). IG decreased junk food consumption (*p* < 0.01) compared to the CG	1b/A
Zask et al. [49] (2012), Australia	RCT	*N* = 1005 children Age 3–6 years *n* CG = 537 *n* IG = 468	To increasing fruit and vegetable intake and decreasing unhealthy food consumption	CG: no intervention IG: “Tooty Fruity Vegie” a game health promotion program	10 months	Dietary intake BMI	**Baseline** *Number of fruit and vegetables serves* CG: 1.95 (0.17) IG: 1.91 (0.13) *BMI z-scores* CG: 0.11 (0.08) IG: 0.14 (0.06) *Mean waist circumference in cm* CG: 52.33 (0.29) IG: 52.54 (0.23) **10 months follow-up** *Number of fruit and vegetables serves* CG: 1.73 (0.12) IG: 2.31 (0.11) *BMI z- scores* CG: 0.24 (0.09) IG: 0.11 (0.06) *Mean waist circumference in cm* CG: 53.49 (0.28) IG: 52.89 (0.29)	1b/A

BMI = body mass index; CG = control group; EL = evidence level; FV = fruit and vegetable; IG = intervention group; RCT = randomized controlled trial; RG = recommendation grade.

## Data Availability

Data available by request to the corresponding author.

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
