# Peer review of "Gamification for the Improvement of Diet, Nutritional Habits, and Body Composition in Children and Adolescents: A Systematic Review and Meta-Analysis"

_nutrients, 2021, doi:10.3390/nu13072478_

Round 1

Reviewer 1 Report

This systematic review and meta-analysis covers an interesting topic that will surely be of increasing importance as time goes on. The systematic review captures 23 studies, 6 of which are meta-analysed. There are major problems with the reporting of this manuscript that should be rectified before it could be suitable for publication, and the manuscript should better reflect parity between its aims, methods and the results presented.

Major comments

  1. In the aims, search strategy and elsewhere, “the outcomes were diet and body composition improvement”. Why were these outcomes not meta-analysed? And, why was nutrition knowledge meta-analysed, even though there was no attention paid to this outcome during the study recovery process (including in the aims and inclusion/exclusion criteria)? These inconsistencies bring the integrity and relevance of the review into doubt, as a search should target the specific outcomes that the review aims to assess.
  2. Meta-analyses should be conducted for all outcomes where at least 2 studies provide suitable data. Data on dietary behaviors could be clustered as appropriate, accounting for differences in stop (sweets, salty snacks, soda) vs start (fruits, vegetables) behaviors. The current presentation of results for food groups, food habits and body composition is unacceptable (vote counting based on significance; See https://training.cochrane.org/handbook/current/chapter-12#section-12-2).
  3. The decision to use a 0-1 transformed version of mean difference as the effect size measure goes against Cochrane guidance on this issue, and has yielded effect size measures that are unrealistically precise (the 95% CI has a range of 0.0 for multiple studies and the cumulative effect). The method used to standardize data on a 0-1 scale is insufficiently described, but it is also probably not appropriate in any case. The mean difference should only be used as an effect measure when all studies assess a construct with the same scale (See https://training.cochrane.org/handbook/current/chapter-10#section-10-5-1). A more appropriate approach would be to use the standardized mean difference (SMD), Cohen’s d or Hedges’ g as an effect size measure, without transforming the data to a 0-1 scale, as this standardization is already done in the process of calculating SMD/d/g.
  4. The cumulative effect size results need to be interpreted/translated into real effects. For example, what does it mean if knowledge increased 0.05? Using SMD/d/g will help with this, as these at least have known quantifications of what constitutes a small, medium or large effect.

Minor comments

  1. Figure 1: Two boxes stats “included in quantitative synthesis”
  2. Section 3.5: The description of results does not match with the forest plot, which shows effects in favor of the control group.
  3. The conclusion “Gamification was an effective method for improving healthy nutritional habits” is wholly unjustified by the data.
  4. Please include a completed PRISMA checklist.

Author Response

Response to Reviewer 1 Comments

Dear Reviewer,

Thank you very much for reviewing the manuscript and your recommendations for improving it. Please find below the response to each recommendation highlighted in yellow. All the changes in the manuscript have also been highlighted in yellow.

This systematic review and meta-analysis cover an interesting topic that will surely be of increasing importance as time goes on. The systematic review captures 23 studies, 6 of which are meta-analyzed. There are major problems with the reporting of this manuscript that should be rectified before it could be suitable for publication, and the manuscript should better reflect parity between its aims, methods and the results presented.

Major comments

Point 1. In the aims, search strategy and elsewhere, “the outcomes were diet and body composition improvement”. Why were these outcomes not meta-analysed? And, why was nutrition knowledge meta-analysed, even though there was no attention paid to this outcome during the study recovery process (including in the aims and inclusion/exclusion criteria)? These inconsistencies bring the integrity and relevance of the review into doubt, as a search should target the specific outcomes that the review aims to assess.

Response 1. Thank you for your analyses. We have meta-analysed the variables with enough data. Nutrition knowledge was assessed because we think that the knowledge is a part of the nutritional habit. We have included the word “knowledge” in the aim and inclusion criteria to avoid that confusion.

Point 2. Meta-analyses should be conducted for all outcomes where at least 2 studies provide suitable data. Data on dietary behaviors could be clustered as appropriate, accounting for differences in stop (sweets, salty snacks, soda) vs start (fruits, vegetables) behaviors. The current presentation of results for food groups, food habits and body composition is unacceptable (vote counting based on significance; See https://training.cochrane.org/handbook/current/chapter-12#section-12-2).

Response 2. Thank you for your comment. We have done the meta-analysis when the data was enough for it and when it was possible to mix the data for a meta-analysis. We have done a new meta-analysis of BMI z-score with 2 studies. The data about vegetables or fruits consumptions in the studies used very different ways of results reporting (calories per day, total grams, % of the food, portions per day, and other types). Thus, we think it is not possible to do the meta-analysis. We have included it as a limitation.  In Table 1, we have included as much statistical information as possible from the studies informing also about the significance, but we have not done vote counting (we have only described the results).

Point 3. The decision to use a 0-1 transformed version of mean difference as the effect size measure goes against Cochrane guidance on this issue, and has yielded effect size measures that are unrealistically precise (the 95% CI has a range of 0.0 for multiple studies and the cumulative effect). The method used to standardize data on a 0-1 scale is insufficiently described, but it is also probably not appropriate in any case. The mean difference should only be used as an effect measure when all studies assess a construct with the same scale (see https://training.cochrane.org/handbook/current/chapter-10#section-10-5-1). A more appropriate approach would be to use the standardized mean difference (SMD), Cohen’s d or Hedges’ g as an effect size measure, without transforming the data to a 0-1 scale, as this standardization is already done in the process of calculating SMD/d/g.

Response 3. The aim of 0-1 transformation is due to the lack of homogeneity among the range of the scales and according to Cochrane's guidance, we performed this approach to erase units and make them comparable. Anyway, we agree with the reviewer that since Cochrane recommends standardized mean deviations and they are immediately implemented by RevMan then. Following your recommendations, we have changed the meta-analysis using the "standardized mean difference" available in the RevMan software from Cochrane. We have modified the forest plot and the results with the new data.

Point 4. The cumulative effect size results need to be interpreted/translated into real effects. For example, what does it mean if knowledge increased 0.05? Using SMD/d/g will help with this, as these at least have known quantifications of what constitutes a small, medium or large effect.

Response 4. We have included more information in the text.

Minor comments

Point 5. Figure 1: Two boxes stats “included in quantitative synthesis”            

Response 5. Thank you for your appreciation. We have changed Figure 1 following the new PRISMA diagram flow and checklist 2020 version.

Point 6. Section 3.5: The description of results does not match with the forest plot, which shows effects in favor of the control group.

Response 6. The standardized means difference, with the 95% confidence interval, was 0.88 (0.05–1.75) displayed a statistically significant higher knowledge score in the values of the experimental group (p <0.05). In the forest plot (figure 2) the means of nutritional knowledge in the experimental group were higher (higher score) than those of the control group (lower score in knowledge).

Point 7. The conclusion “Gamification was an effective method for improving healthy nutritional habits” is wholly unjustified by the data.

Response 7. Thank you for your appreciation. We have modified the conclusion.

Point 8. Please include a completed PRISMA checklist.

Response 8. Thanks for the recommendation. We have included the PRISMA 2020 checklist.

Reviewer 2 Report

Well done.

My only suggestion is in the abstract.    Currently, one of the main public health problems among children and adolescents is poor adherence to healthy habits, leading to increasingly high rates of obesity and the comorbidities that accompany obesitynon-communicable
diseases.

Author Response

Response to Reviewer 2 Comments

Dear Reviewer,

Thank you very much for reviewing the manuscript and your recommendations for improving it. Please find below the response to each recommendation highlighted in yellow. All the changes in the manuscript have also been highlighted in yellow.

Point 1: My only suggestion is in the abstract. “Currently, one of the main public health problems among children and adolescents is poor adherence to healthy habits, leading to increasingly high rates of obesity and the comorbidities that accompany obesity”.

Response 1: Thanks for the suggestion. We have modified the abstract.